Automated lung cancer diagnosis from chest X-ray images using convolutional neural networks

Aboelghiet Aya 1
http://orcid.org/0000-0002-2832-6171 Shohieb Samaa M. 1 sm_shohieb@mans.edu.eg
http://orcid.org/0000-0002-7646-4915 Rezk Amira 1
Abou Elfetouh Ahmed 1 2
Sharaf Ahmed 3 aisharaf@uqu.edu.sa
Abdelmaksoud Islam 1
1 Department of Information Systems, Faculty of Computers and Information, Mansoura University , Mansoura, Daqahlia , Egypt
2 Delta Higher Institute for Management and Accounting Information Systems, Delta Higher Institute for Management and Accounting Information Systems , Mansoura, Daqahlia , Egypt
3 Deanship of Postgraduate Studies And Research, Umm Al-Qura University , Makkah , Saudi Arabia
Wan Shibiao
Electronic publication date: 2025 Sep 4
Publication date: 2025
Volume: 11
Electronic Location ID: e3145
Received 2025 Mar 10; Accepted 2025 Jul 30
Copyright: © 2025 Aboelghiet et al.
Copyright year: 2025
Copyright holder: Aboelghiet et al.
License: This is an open access article distributed under the terms of the Creative Commons Attribution License, which permits unrestricted use, distribution, reproduction and adaptation in any medium and for any purpose provided that it is properly attributed. For attribution, the original author(s), title, publication source (PeerJ Computer Science) and either DOI or URL of the article must be cited.
License URL: https://creativecommons.org/licenses/by/4.0/

Keywords: Lung cancer, Chest X-ray, Computer-aided diagnosis, Convolutional neural networks, CNN

Funding: Umm AL-Qura University, Saudi Arabia 25UQU4330113GSSR01 This research work was funded by Umm AL-Qura University, Saudi Arabia under grant number: 25UQU4330113GSSR01. The funders had no role in study design, data collection and analysis, decision to publish, or preparation of the manuscript.

==============================
Background/Objectives

Lung cancer is the leading cause of cancer-related deaths worldwide. While computed tomography (CT) scans provide more comprehensive medical information than chest X-rays (CXR), the high cost and limited availability of CT technology in rural areas pose significant challenges. CXR images, however, could serve as a potential preliminary diagnostic tool in diagnosing lung cancer, especially when combined with a computer-aided diagnosis (CAD) system. This study aims to enhance the accuracy and accessibility of lung cancer detection using a custom-designed convolutional neural network (CNN) trained on CXR images.

Methods

A custom-designed CNN was trained on an openly accessible CXR dataset from the Japanese Society for Radiological Technology (JSRT). Prior to training, the dataset underwent preprocessing, where each image was divided into overlapping patches. A t-test was applied to these patches to distinguish relevant from irrelevant ones. The relevant patches were retained for training the CNN model, while the irrelevant patches were excluded to enhance the model’s performance.

Results

The proposed model yielded a mean accuracy of 83.2 ± 2.91%, demonstrating its potential as a cost-effective and accessible preliminary diagnostic tool for lung cancer.

Conclusions

This approach could significantly improve the accuracy and accessibility of lung cancer detection, making it a viable option in resource-limited settings.

Introduction

Lung cancer is the second most common type of cancer worldwide. According to World Cancer Statistics 2022, lung cancer accounts for 2.5 million new instances and 18.7% of the total cancer deaths (WHO, 2024). Lung cancer is characterized by uncontrolled growth of abnormal cells in the lung tissue (Abunajm et al., 2023). Two types of lung cancer exist: small-cell lung cancer (SCLC) and non-small cell lung cancer (NSCLC), with NSCLC being the most common, accounting for 85% of all lung cancer cases (Tashtoush et al., 2023). Lung cancer is often caused the synergy between carcinogen exposure, smoking, genetic factors, and air pollution of varying natures (Agonsanou, Figueiredo & Bergeron, 2023).

Lung cancer tends not to exhibit any symptoms or signs in its initial phase. However, the availability of automatic diagnosis systems for lung cancer promotes early detection, thereby enhancing the chances of effective treatment and cure rate by 20% (Huang et al., 2023). Lung cancer is diagnosed by sputum examination, lung examination (bronchoscopy and lung tissue biopsy), and medical imaging (Demirci, 2023). Sputum examination is unable to detect small adenocarcinomas of less than 2 cm in diameter and is insensitive to screen the suspected population with lung cancer (Lee et al., 2023). The bronchoscopist’s skill plays an important role in the right diagnosis during lung check-ups. Although bronchoscopy is a provocative procedure, it may bring uneasiness to patients and have complications (Kavas & Yildiz, 2023). Limitations of sputum and lung examination include cost, application of anesthesia, and the chance of infection, which indicate that there is a need for uncomplicated and cheap methods of identifying lung cancer. Recent studies show that analysis of imaging modalities achieves better results than sputum analysis and chest examination (Hua et al., 2023). The most common imaging modalities used in the diagnosis of lung diseases are chest x-rays (CXR), computed tomography (CT), and magnetic resonance imaging (MRI) (Daniel, Cenggoro & Pardamean, 2023). Each of these imaging modalities has their advantages and drawbacks. For example, the advantages of CXR images include cost-effectiveness and availability in rural areas. On the other hand, CXR images suffer from low quality and noise (Abd-ElGhaffar et al., 2022). Deep learning (DL)-based computer-aided diagnosis (CAD) systems that employ convolutional neural networks (CNNs) can provide accurate diagnosis of lung cancer. In this article, a CAD system for detecting lung cancer using a custom-designed CNN is proposed. This custom-designed CNN is trained on a publicly-available CXR dataset collected through the Japanese Society for Radiological Technology (JSRT). The dataset is preprocessed by splitting each CXR image into a set of overlapping patches. Then the patches are processed using a t-test to identify useful patches and discard useless ones in order to enhance the efficiency of the CNN specifically designed. The contributions of this article are outlined below: This article proposes a custom-designed CNN model for lung cancer diagnosis from CXR images. The model is specially designed for the task and is therefore superior to existing models with regards to accuracy and robustness.

The article presents a new method to solve the issue of small training datasets by splitting CXR images into overlapping patches. Splitting of each CXR image into a set of overlapping patches increases the number of training samples and handles the overfitting problem common with small datasets.

The article demonstrates a new method of preprocessing by applying a t-test to the generated patches to distinguish between relevant and irrelevant ones. This ensures that the most informative patches are used during training, which can enhance the accuracy of the CNN model.

The proposed model is particularly valuable in resource-limited settings where access to advanced imaging technologies is constrained.

The rest of the article is organized as follows: ‘Literature Review’ provides a review of the literature on lung cancer diagnosis. ‘Proposed Diagnosis System’ presents the main processing steps of the proposed diagnostic system, the architectures of the employed CNNs, and a comparison with three well-known CNNs, AlexNet, GoogLeNet, and ResNet-34. ‘Experiment Results and Discussion’ discusses the experiment results of the proposed diagnostic system. Finally, ‘Model Results’ concludes the article.

Literature review

Over the past few decades, many studies have used artificial intelligence (AI) in detecting lung cancer through medical imaging. Among all the imaging techniques in medicine, CT scan stands out the most prominent in lung cancer diagnosis since it is extremely sensitive in detecting nodules otherwise not easily visible, thus boosting the accuracy of diagnosis. However, it has harmful X-ray radiation, has a lot of processing time, and is hardly accessible in remote rural areas far from towns due to its very high cost. All these drawbacks can be eliminated through the utilization of CXR. CXR imaging is traditionally an old modality of medical imaging. But the newly emerged improvement in machine learning (ML) has enhanced the role of CXR as an imaging modality for diagnosis. For example, Joon, Bajaj & Jatain (2019) detection of lung cancer from CXR images using support vector machine (SVM) algorithm. The approach started pre-processing of CXR images using denoising clustering algorithms, feature extraction from the segmented image using K-means and fuzzy C-means. Afterwards, classification was done using the SVM-based model whether the CXR image has lung cancer or not. The traditional ML techniques for the detection of lung cancer also possess certain limitations: Handcrafted feature extraction, which can be used for the identification of input images with accuracy, is not easy as it requires field experience. DL techniques, however, can extract discriminatory features of input images automatically, which can potentially identify and detect lung nodules with accuracy. Several studies employed DL for lung cancer classification. For example, Mendoza & Pedrini (2020) suggested a CAD system consisting of lung segmentation, nodule localization, and nodule candidate classification to detect lung nodules from CXR images. Their performance depended on the choice of data augmentation parameters and designing a specific CNN architecture. Their experiments showed that CNNs can perform as well as high-accuracy approaches. Horry et al. (2022) have explained that detection of lung cancer from CXR images can be improved by applying a preprocessing pipeline to remove brightness and contrast variations in CXR images using histogram equalization. Model training was conducted through combinations of different lung field segmentation, rib suppression operators, and close cropping. Several classification schemes for lung cancer have been introduced on the JSRT database, e.g., Rajagopalan & Babu (2020) classified lung cancer in the JRST dataset employing a massive artificial neural network (MANN) with a soft tissue approach. The proposed CAD system’s accuracy was 72.96%. Uçar & Uçar (2019) proposed a CNN-based model that preprocessed input images using the Laplacian of a Gaussian (LoG) filter. The diagnostic accuracy reached 82.43% when LoG filter was used whereas the accuracy of the model was 72.9% when LoG filter was not used. Ausawalaithong et al. (2018) proposed a DenseNet-121-based model that achieved an accuracy of 74.43%. In this study, AlexNet and GoogLeNet models have been employed as baseline architectures. These have been selected because of their state-of-the-art performance in various image classification tasks. Both architectures have been extensively used in medical imaging studies and have been thoroughly studied. Their inclusion offers a relevant benchmark against which to measure the success of our proposed approach. In the medical imaging literature, AlexNet is frequently used as a baseline model, facilitating straightforward comparison with prior research on classifying chest X-rays (Krizhevsky, Sutskever & Hinton, 2012). Its shallow architecture, comprising only eight layers, is simpler to interpret and minimizes the risk of overfitting small sets of data. GoogLeNet integrates multi-scale convolutional filters (1 × 1, 3 × 3, 5 × 5) within parallel pathways in order to improve feature extraction for patterns. Although, it consists of 22 layers, parameter sparsity (achieved via 1 × 1 convolutions) ensures computational efficiency while maintaining diagnostic accuracy. Previous studies have demonstrated that the accuracy of lung cancer diagnosis using deep models such as CNNs is highly affected by the small size of the training data. To address this limitation, the proposed model generates overlapping patches from CXR images by dividing each input image into smaller sub-images. These sub-images are then utilized to train the developed CNN to increase the size of the training dataset. Irrelevant patches are then excluded by applying a t-test in order to retain only the most informative patches to train the CNN model to improve its performance.

Proposed diagnosis system

In this article, a CAD system is proposed to detect lung cancer in CXR images using CNNs. The proposed CAD system consists of three main steps: In the first step, the input CXR images are preprocessed by dividing each image into a set of overlapping patches. Then, a t-test is applied on these patches to retain the relevant patches and exclude irrelevant ones. In the second step, a custom-designed CNN is used to automatically extract discriminatory features from the image patches. In the final step, a softmax classifier utilizes the learned feature from the second step to determine whether the patches are cancerous or normal. These steps are illustrated in Fig. 1.

Figure 1 An illustration of the proposed framework for diagnosing lung cancer from CXR images.

Data preprocessing

CXR images are divided into overlapping patches to expand the dataset for CNN training. Irrelevant patches are excluded using a t-test to improve performance.

Patch generation

To address the small dataset size (247 CXR images), each image is divided into a set of patches. There are two types of patches: adjacent patches and overlapping patches. Overlapping patches, which predict each pixel based on its neighbors, are shown to outperform adjacent patches in accuracy (Ben naceur et al., 2020). Experiments were carried out to select the optimal number of patches per image to obtain the optimal compromise among training time, computational capabilities, and the model’s accuracy, and it was found that 11 patches per image are sufficient for the efficient training of CNNs. It was tried using configurations involving eight, 11, and 15 patches. Reduced patch count (eight) brought the accuracy to 82.47%, because with this quantity of patches it could not manage to cover up all the CXR images. On the other hand, 15 patches incurred computational overhead without achieving increases in accuracy. Configuration 11 offered 83.19% classification accuracy and is therefore a compromise between capturing significant parts of the image and computationally tractable complexity. The performance of the CNN model under different patch numbers is summarized in Table 1.

Table 1 Performance metrics of the proposed CNN model after varying numbers of patches.

Num patches	Accuracy (%)	Sensitivity (%)	Specificity (%)	Precision (%)	F1-score (%)	
8	82.47 ± 1.14	88.2 ± 2.3	79.1 ± 2.6	72.7 ± 4.3	79.37 ± 2.9	
11	83.19 ± 2.9	83.7 ± 4.5	82.8 ± 5.9	75.3 ± 5.8	79.4 ± 2.7	
15	83.13 ± 18.5	98.0 ± 6.3	73.9 ± 31.8	76.6 ± 20.8	84.12 ± 13.6	

In this study, the effect of modifying the overlap ratio between patches on the performance of a model was empirically investigated. Specifically, model performance was measured according to overlap ratios of less than 25% and 25%. Increased overlap ratio from less than 25% to 25% was found to produce moderate improvement in classification accuracy from 83.2% to 83.7%. This improvement in performance is attributed to the added contextual information brought with higher overlap, which can potentially identify slight abnormalities such as small nodules. Nevertheless, it was observed that there exists a region of overlap where high overlap can add redundancy and complexity to the computations without resulting in useful improvement in performance. Based on such observations, the overlap ratio of 25% is considered an optimal compromise between model performance and computational expense.

t-test analysis

After the input images are divided into a set of overlapping patches, some patches are composed entirely or mostly of non-lung areas. To remove irrelevant patches, a t-test is applied to the overlapping patches, focusing on lung areas. Experiments revealed that patches with a p-value lower 0.003 are retained, yielding optimal classification results. A t-test is applied to compare pixel intensities between the lung and non-lung regions within each patch. The lung region represents group 1, while the non-lung region represents group 2. For each patch, the mean pixel intensity of both regions is calculated and a t-test is applied to determine whether the difference in means is statistically significant. The null hypothesis of the t-test assumes that there is no difference between the lung and non-lung regions. If the p-value calculated from the t-test is less than a predefined threshold (p < 0.003), the patch is considered relevant and retained for training. Otherwise, the patch is considered irrelevant and excluded. Different threshold values were tested to identify a value that balances accuracy and computational efficiency.

Biologically, lung cancer often manifests as localized lesions or masses that show significant intensity differences compared to healthy lung tissue. The t-test effectively identifies patches where such differences are statistically significant, which indicates areas where abnormalities may be present. Moreover, radiologists focus on the lung region when diagnosing lung cancer. Abnormalities in this region, such as nodules or masses, are key indicators of the disease. By using the t-test to isolate relevant lung regions, our approach aligns with clinical practice, which allows the model to focus on the regions most likely to exhibit signs of disease.

The patch-based method, combined with the t-test, offers a more targeted technique compared to other common data augmentation methods such as rotation, flipping, or scaling, which might introduce variations less directly relevant to lung cancer detection. With the use of patches and the t-test, spatial integrity of critical features is preserved while addressing the problem of small datasets, allowing the model to acquire improved accuracy.

Discriminatory features extraction

A CNN consists mainly of three layer types: convolutional (Conv) layers, pooling layers, and fully connected (FC) layers. The Conv and pooling layers are responsible for feature extraction, while the FC layers map these extracted features into the final output.

Developed CNN architecture

A customized CNN is designed to classify lung images, as illustrated in Fig. 2. This framework consists of five blocks, each with three layers: a two-dimensional convolution (Conv2D) layer, a batch normalization (BN) layer, and a dropout layer. Following these blocks are a pooling layer, a flatten layer, a fusion layer, a FC layer, and a softmax layer. A brief description of these components is given below:

Conv2D layer: The model begins with the Conv2D layer, which extracts features. A 256 × 256 patch size is used, convolved with a 2 × 2 filter matrix, and the rectified linear unit (ReLU) function is applied to transfer the output to the next layer.

BN layer: Training CNNs can be challenging, partly because the input from earlier layers may vary after weight updates. BN is used to standardize the model’s input. Additionally, this layer reduces training time and provides the model with some regularization.

Dropout layer: This layer mitigates the problem of overfitting the model. In our model, dropout layers are applied with a rate of 0.5.

Pooling layer: This layer reduces the feature map size while preserving essential information. This helps increase computation speed, reduce memory requirements, and control overfitting. Our model employs average pooling.

Flatten layer: This layer converts the 2D convolutions into a 1D array for the next layer.

FC layer: Also known as a dense layer, it is used in the final stages of CNN to classify input patches.

Figure 2 An illustration of the developed CNN architecture for diagnosing of lung cancer in overlapping patches.

Comparison with AlexNet, GoogLeNet, and ResNet-34

The performance of the proposed CNN model was evaluated against three well-known CNN architectures: AlexNet, GoogLeNet, and ResNet-34. AlexNet comprises eight layers: five Conv2D layers and three FC layers. GoogLeNet, based on the Inception architecture, captures multi-scale features by applying filters of different sizes to the same input and concatenating the results. However, it can lose important information as it reduces the size of the feature maps. Its architecture includes 22 layers, consisting of Conv, pooling, and FC layers. GoogLeNet was developed to improve the accuracy over earlier neural networks (Mahmmed & Majeed, 2023).

ResNet-34 is a neural network architecture that consists of 34 layers and employs residual connections (skip connections) to improve the training of deep neural networks and mitigate the vanishing gradient problem. A comparison of the AlexNet, GoogLeNet, and ResNet-34 architectures is shown in Table 2.

Table 2 Differences between proposed CNN model, AlexNet, GoogleNet, and ResNet-34.

Features	Proposed CNN model	AlexNet	GoogleNet	ResNet-34	
Architecture deep	(16 layers)	(8 layers)	(22 layers)	(34 layers)	
Activation function	ReLU	ReLU	ReLU	ReLU	
Pooling	Non-overlapping	Overlapping	Non-overlapping	Non-overlapping	
Convolution	Parallel convolution	Consecutive	Parallel (inception)	Sequential convolutions	
Dimensionality	Convolutions	No reduction	1 × 1 Convolution	Convolutions	
Regularization	Dropout	Dropout	Auxiliary Classifiers	Batch normalization	

Experiment results and discussion

Dataset description

A publicly available standard digital image dataset, collected by the JSRT, is used (El Mansouri, El Mourabit & El Habouz, 2022). The JSRT dataset comprises 247 CXR images. Radiologists examined these images and identified 154 instances of lung cancer and 93 instances without lung cancer (Ausawalaithong et al., 2018).

Performance metrics

Multiple metrics are used to evaluate classification models. These metrics are calculated using the values of a confusion matrix. The confusion matrix has four factors: true positive (TP), true negative (TN), false positive (FP), and false negative (FN). Using these values, the following performance metrics are derived: a) Accuracy: The accuracy of a model represents its ability to differentiate the patient and healthy cases correctly. Accuracy can be computed using the following formula (Maxwell, Warner & Guillén, 2021): (1) Accuracy=TP+TNTP+TN+FP+FN

b) Sensitivity (recall): The sensitivity of a model represents its ability to identify the patient cases correctly. Sensitivity can be computed using the following formula (Agarwal et al., 2021): (2) Sensitivity(Recall)=TPTP+FN

c) Specificity: The specificity of a model represents its ability to identify the healthy cases correctly. Specificity can be computed using the following formula (Miao & Zhu, 2022): (3) Specificity=TNTN+FP

d) Precision: It is the ratio of true positives and total positives predicted. Precision can be computed using the following formula (Niu et al., 2022): (4) Precision=TPTP+FP

e) F1-score: A combined measure that includes both precision and recall as it is the harmonic mean of the two. F1-score can be computed using the following formula (Yacouby & Axman, 2020): (5) F1-score=2×Precision×RecallPrecision+Recall.

Model results

An experiment was conducted to evaluate the performance of the developed CNN model using 10-fold cross-validation. This method accurately reflects the generalization ability of the evaluated model. In 10-fold cross-validation, the dataset is divided into ten subsets. The model is trained on nine subsets and validated on the remaining subset. This process is repeated 10 times until each subset has been used as a validation set. The results of cross-validation are more robust as they reflect the performance across the different subsets. The obtained performance metrics of the custom-designed CNN model were recorded for various epoch counts in order to show how the performance evolved with different durations. The proposed custom-designed CNN achieved the lowest accuracy when the number of epochs was set to 50. However, the accuracy improved as the number of epochs increased. Table 3 shows the obtained performance measures of the custom-designed CNN model for different number of epochs. The results suggest that increasing the number of epochs generally improves the model’s performance, especially in terms of sensitivity and accuracy. Optimal performance is achieved with a specific number of epochs, typically around 90 to 100, beyond which the performance gains become marginal.

Table 3 Performance metrics of the proposed CNN model after varying numbers of epochs.

Epoch	Accuracy (%)	Sensitivity (%)	Specificity (%)	Precision (%)	F1-score (%)	
50	81.9 ± 2.36	75.59 ± 4.9	85.7 ± 6.2	77.1 ± 6.8	75.9 ± 1.8	
70	82.14 ± 1.5	77.8 ± 7.65	84.7 ± 5.85	76.38 ± 6.1	76.5 ± 2.05	
90	83.1 ± 2.4	80.3 ± 4.2	84.7 ± 6.05	76.7 ± 6.3	78.2 ± 1.9	
100	83.19 ± 2.9	83.7 ± 4.5	82.8 ± 5.9	75.3 ± 5.8	79.4 ± 2.7	

Figure 3 illustrates the average performance metrics derived from the values listed in Table 3 across various epochs, demonstrating the model’s performance over time.

Figure 3 The evaluated performance metrics analysis for proposed custom-designed CNN model.

To illustrate the impact of the t-test on the custom CNN’s performance, an experiment was conducted by training the model with and without the t-test applied to the input patches. Table 4 compares the results, showing that the t-test improves model performance.

Table 4 Results of training the CNN model with and without applying the t-test.

Model	Accuracy (%)	Sensitivity (%)	Specificity (%)	Precision (%)	F1-score (%)	
CNN with t-test	83.19 ± 2.9	83.7 ± 4.5	82.8 ± 5.9	75.3 ± 5.8	79.4 ± 2.7	
CNN without t-test	69.9 ± 2.36	63.46 ± 1.6	71.3 ± 3.18	65.25 ± 6.8	63.17 ± 2.7	

Figure 4 presents a bar chart of the performance metrics from Table 4, comparing the custom-designed CNN model with and without the t-test. This visual representation provides a clearer comparison of the model’s performance under both conditions.

Figure 4 The evaluated performance metrics for comparison of used t-test.

Additional experiments were conducted to compare the performance of the developed CNN model with two well-known CNN architectures, namely AlexNet and GoogLeNet. The objective was to assess how the developed model performs against established benchmarks. The comparative experimental outcomes reveal that the suggested CNN model performed best out of the three models. Quantitative performance results are presented in Table 5. Our comparative work is supplemented by reporting outcomes on an additional architecture: ResNet-34. It is a newer and deeper architecture using residual connections to minimize vanishing gradient issues. ResNet-34 consists of 34 layers and is designed to enhance feature extraction in deep networks. The result of this extended comparison has been added to Table 5. ResNet-34 offers higher accuracy compared to AlexNet and GoogleNet. This comparison indicates the effectiveness of the custom CNN with its unique design in offering improved performance in lung cancer detection.

Table 5 Performance comparison of proposed CNN model, ResNet-34, GoogLeNet, and AlexNet using two training strategies: full image vs. t-test selected patches.

Model	Training strategy full image/Patches (with t-test)	Accuracy (%)	Sensitivity (%)	Specificity (%)	Precision (%)	F1-score (%)	
Proposed CNN model	Full image	84.44 ± 3.5	82.09 ± 5.4	85.93 ±3.2	83.06 ± 4.3	81.27 ± 2.3	
Proposed CNN model	Patches	83.19 ± 2.9	83.7 ± 4.5	82.8 ± 5.9	75.3 ± 5.8	79.4 ± 2.7	
ResNet-34	Full image	73.2 ± 27.1	84.5 ± 23.9	66.25 ± 37.3	67.93 ± 31.9	73.3 ± 27.3	
ResNet-34	Patches	76.5 ± 11.7	93.9 ± 13.9	65.9 ± 13.5	63.24 ± 12.3	75.3 ± 12.3	
GoogLeNet	Full image	66.2 ± 19.2	67.43 ± 34.7	64.75 ± 25.7	54.4 ± 24.8	58.2 ± 27.1	
GoogLeNet	Patches	71.82 ± 1.3	79.42 ± 3.9	70.25 ± 4.18	69.4 ± 1.14	74.2 ± 3.6	
AlexNet	Full image	64.85 ± 20.6	76.78 ± 26.8	57.35 ± 28.9	55.31 ± 23.4	62.67 ± 22.6	
AlexNet	Patches	67.62 ± 2.14	78.13 ± 2.7	76.6 ± 1.8	60.46 ± 1.4	68.2 ± 2.5	

To ensure a fair comparison focused on architectural differences, all models in Table 5 were trained and evaluated using the same t-test-selected patches. The proposed CNN consistently outperformed ResNet-34, GoogLeNet, and AlexNet under these conditions. This indicates that its architectural modifications enhance performance. To further clarify the source of improvement, Table 5 also includes results for training on the original full image without t-test preprocessing. The comparison shows that t-test-based patch selection generally improves performance and reduces variability for ResNet-34, GoogLeNet, and AlexNet. This highlights the t-test-based patch selection enhances both accuracy and model stability. Although the proposed CNN performed lightly better on full images than on t-test patches, the difference is marginal and within the standard deviation.

As reported in Table 5, the proposed model trained on full images achieved slightly higher accuracy (84.44%) compared to training on patches (83.19%). While full-image training might be slightly superior in performance when data is sufficient, the patch-based strategy offers better generalization in limited-data scenarios. The patch-based strategy was primarily introduced to address the issue of limited dataset size and to prevent overfitting during training. Additionally, full-image training demands extensive graphics processing unit (GPU) memory and significantly longer training times, whereas patch-based training reduces memory requirements, which enables the model to be trained on standard hardware. The patch-based approach, especially when combined with the t-test, increases the effective number of training samples without relying on image augmentation. Moreover, training on patches enables the model to focus on localized abnormalities, which are clinically important for early-stage lung cancer detection. Full image training may dilute subtle lesion features due to the dominance of healthy tissues, whereas patches allow the model to focus on abnormal regions.

Thresholding is a simpler and commonly used technique for localizing lung regions. To evaluate its effectiveness relative to the employed t-test-based patch selection, an additional experiment was conducted using adaptive thresholding based on pixel intensity. The results of that experiment are presented in Table 6. As shown, the t-test approach outperformed thresholding across all performance metrics. These results suggest that while thresholding can offer a quick approximation, the t-test method more effectively identifies discriminative regions relevant for classification, which leads to improved overall performance.

Table 6 Performance comparison: patches (t-test) vs. thresholding approaches.

Image preprocessing method	Accuracy (%)	Sensitivity (%)	Specificity (%)	Precision (%)	F1-score (%)	
Patches (with t-test)	83.19 ± 2.9	83.7 ± 4.5	82.8 ± 5.9	75.3 ± 5.8	79.4 ± 2.7	
Thresholding	79.21 ± 1.6	79.42 ± 1.3	70.25 ± 2.13	69.4 ± 2.4	74.2 ± 2.1	

To highlight the efficiency of the proposed model compared to other models, a comparative comparison was done using other models available in the literature that were implemented for the same dataset. All performance metrics outperformed the new CNN model compared to other methods. The approaches applied in Li et al. (2022), Rajagopalan & Babu (2020), and Ausawalaithong et al. (2018) yielded the lowest accuracies of 73.92%, 72.96%, and 74.43%, respectively, while research in Uçar & Uçar (2019) yielded a moderate accuracy of 82.43%. Nevertheless, the created model yielded the highest accuracy of 83.19%. The results of this comparison are presented in Table 7. This comparative evaluation highlights that the performance of custom-designed CNN is improved after applying the proposed preprocessing techniques, such as dividing the CXR images into overlapping patches and applying a t-test to retain only the most significant. In order to make it clinically useful, the model’s diagnostic accuracy is evaluated at the full-image level because clinical diagnosis relies on global assessment of whole chest X-rays and not patches.

Table 7 Comparative analysis with previous research.

Study	Method type (Patches vs. Full images)	Accuracy
(%)	Sensitivity
(%)	Specificity
(%)	Precision
(%)	F1-score
(%)	
Proposed CNN model	Patches	83.19	83.7	82.8	75.3	79.4	
Proposed CNN model	Full images	84.44	82.09	85.93	83.06	81.27	
Li et al. (2022)	Full images	73.92	72.42	–	75.2	72.92	
Rajagopalan & Babu (2020)	Full images	72.96	72.85	–	–	–	
Uçar & Uçar (2019)	Full images	82.43	–	–	–	–	
Ausawalaithong et al. (2018)	Full images	74.43	74.68	74.96	–	–	

Conclusions

In this study, a CAD system is proposed for lung cancer detection from CXR images using a CNN. The CAD system is trained using the JSRT dataset. One significant challenge is the small size of the JSRT dataset. If this dataset is used to train deep architectures such as CNNs, this can lead to the problem of overfitting. To address this problem, each input CXR image of the JSRT dataset was divided into a set of overlapping patches to increase the size of the training dataset. Then, a t-test was applied to these patches to retain only the most informative patches and exclude irrelevant ones. The proposed CNN model achieved an accuracy of 83.19%, demonstrating its superiority over existing models in comparative analysis. This model is particularly valuable in resource-limited environments, as it enables more affordable and lower-risk lung cancer screening using CXR images. For future work, the application of image data augmentation techniques, including color space adjustments, feature space augmentation, kernel filters, and the incorporation of additional features such as family history, smoking rate could enhance the accuracy of CAD system for diagnosis CXR image. Datasets from different populations, such as the NIH Chest X-ray dataset and the MIMIC-CXR dataset, will be incorporated. These datasets include not only lung cancer cases but also multiple other diseases, which will provide a more comprehensive evaluation of the model’s performance across a wider range of conditions.

Supplemental Information

Supplemental Information 1 Code.

Updated to ensure that it runs following recent updates to certain libraries.

Supplemental Information 2 Source code.

Additional Information and Declarations

Competing Interests

The authors declare that they have no competing interests.

Author Contributions

Aya Aboelghiet conceived and designed the experiments, performed the experiments, analyzed the data, performed the computation work, prepared figures and/or tables, authored or reviewed drafts of the article, and approved the final draft.

Samaa M. Shohieb conceived and designed the experiments, performed the experiments, analyzed the data, performed the computation work, prepared figures and/or tables, authored or reviewed drafts of the article, and approved the final draft.

Amira Rezk conceived and designed the experiments, performed the experiments, analyzed the data, performed the computation work, prepared figures and/or tables, authored or reviewed drafts of the article, and approved the final draft.

Ahmed Abou Elfetouh conceived and designed the experiments, performed the experiments, analyzed the data, performed the computation work, prepared figures and/or tables, authored or reviewed drafts of the article, and approved the final draft.

Ahmed Sharaf conceived and designed the experiments, performed the experiments, analyzed the data, performed the computation work, prepared figures and/or tables, authored or reviewed drafts of the article, and approved the final draft.

Islam Abdelmaksoud conceived and designed the experiments, performed the experiments, analyzed the data, performed the computation work, prepared figures and/or tables, authored or reviewed drafts of the article, and approved the final draft.

Data Availability

The following information was supplied regarding data availability:

The The Japanese Society of Radiological Technology (JSRT) dataset is available at: http://db.jsrt.or.jp/eng.php, https://www.kaggle.com/datasets/raddar/nodules-in-chest-xrays-jsrt.

Source code is available at GitHub and Zenodo:

https://github.com/Aya20989/All-Source-Code.git.

aya111166. (2025). aya111166/source-code: source code (1.0). Zenodo. https://doi.org/10.5281/zenodo.15737582.

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
