# Peer review of "Automated lung cancer diagnosis from chest X-ray images using convolutional neural networks"

_PeerJ Computer Science, doi:10.7717/peerj-cs.3145_

## Round 0.1 · original submission · Major Revisions

The reviewers have substantial concerns about this manuscript. The authors should provide point-to-point responses to address all the concerns and provide a revised manuscript with the revised parts being marked in different color.

Reviewer 1 ·

Basic reporting

no comment

Experimental design

The manuscript "Automated Lung Cancer Diagnosis from Chest X-ray Images Using Convolutional Neural Networks" presents a method for lung cancer detection using chest X-ray images. Overall, it provides a clear description of the research question and relevant background information. However, the validity of the findings requires significant improvement, and the methodology needs further clarification. Below are the detailed comments.
1. In the patch generation section, the authors state that "experiments revealed that 11 patches per image were sufficient for effective CNN training." However, no supporting evidence is provided in the results section. The authors should present justification for choosing 11 patches over other possible numbers, such as 4, 8, or 12, to substantiate this claim.
2. In the T-test analysis section, the authors state that "to remove irrelevant patches, a t-test is applied to the overlapping patches, focusing on lung areas." However, it is unclear how the comparison was performed and how the p-value was calculated. The authors should provide further clarification on the methodology used in this analysis.
3. Figure 1 shows that the model outputs a classification for each patch, indicating whether it is normal or cancerous. However, in clinical practice, diagnosis is based on the entire chest X-ray rather than individual patches. Therefore, it may also be necessary to evaluate the model’s prediction accuracy at the full-image level and compare its performance with existing models.
4. The authors compare the proposed method with existing models such as GoogleNet and AlexNet. It would be beneficial for them to justify the choice of these models. Additionally, consideration of other architectures, such as ResNet-18 or ResNet-34, could further strengthen the comparison.

Validity of the findings

5. Following the previous comment, the authors state that “the work in Uçar and Uçar (2019) achieved a moderate accuracy of 82.43%.” The author also provided comparisons with previous research in Table 5. However, it appears that Uçar’s study evaluates the model using whole images rather than individual patches. As a result, the findings of the current manuscript are not directly comparable to those of Uçar’s work. Additionally, the authors should carefully review the other methods to determine whether they are directly comparable.
6. Since the authors use 10-fold cross-validation, it would be beneficial to include error bars in Figures 3 and 4 to illustrate the variance across different folds.
7. In table 1, The author can also include their own model to provide a clearer comparison.
8. I appreciate that the author provided the code both on GitHub and in the supplemental materials. However, I encountered issues when attempting to open the Jupyter notebook in Jupyter; neither version was accessible. I recommend the author double-check the uploaded code to ensure it is functioning as expected.

Reviewer 2 ·

Basic reporting

1, Please provide valid source code.
When I click the link of code https://github.com/Aya20989/github provided by the author, the page shows: "Invalid Notebook
The Notebook Does Not Appear to Be Valid JSON".
I downloaded the code file and tried to open it on my computer by jupyter notebook, it shows:
"File Load Error for Computer Code.ipynb. Unreadable Notebook."

Experimental design

1, The main novelty of the method part is the t-test in data preprocessing as described below:
"To remove irrelevant patches, a t-test is applied to the overlapping patches, focusing on lung areas. Experiments revealed that patches with a p-value of 0.003 or lower are retained, yielding optimal classification results."
However, how is t-test applied is not clear stated. t test should use two groups, please specify the groups and values.

Validity of the findings

1, In Differences between AlexNet and GoogleNet Table, please also include the proposed model.

Reviewer 3 ·

Basic reporting

This manuscript presents an automated diagnostic system for lung cancer detection from chest X-ray (CXR) images using a custom-designed convolutional neural network (CNN). The authors propose a novel preprocessing method involving dividing CXR images into overlapping patches and employing a t-test to retain informative patches, thus addressing dataset size limitations. The developed CNN is compared with established models, AlexNet and GoogLeNet, achieving superior accuracy (83.19%). The author did not clearly explain the rationale for using a patch strategy and t-test to enhance the datasets compared to other augmentation strategies. Further explanation is required.

Experimental design

1. How does the overlap ratio of patches impact model performance?
2. Why did the author choose to use patch images instead of other data augmentation strategies like rotation, noise addition, or scaling?

Validity of the findings

3. How did the author conduct the t-test? The authors could clarify its biological or clinical relevance further. Specifically, how does the t-test effectively distinguish relevant lung regions?
4. The manuscript would benefit from evaluating performance on additional datasets to assess generalizability and robustness across diverse populations and imaging conditions.
5. The literature review provides sufficient context but lacks detailed justification for choosing CNN architectures (AlexNet and GoogLeNet) specifically for comparative analysis.

Additional comments

Figures are a little bit blurred, please revise them accordingly.

---

## Round 0.2 · Major Revisions

There are still some remaining major concerns that need to be addressed.

Reviewer 1 ·

Basic reporting

The authors have addressed all my comments.

Experimental design

The authors have addressed all my comments.

Validity of the findings

The authors have addressed all my comments.

Additional comments

The authors have addressed all my comments.

Reviewer 3 ·

Basic reporting

Summary: This manuscript presents an automated diagnostic system for lung cancer detection from chest X-ray (CXR) images using a custom-designed convolutional neural network (CNN). The authors propose a novel preprocessing method involving dividing CXR images into overlapping patches and employing a t-test to retain informative patches, thus addressing dataset size limitations. The author revised the manuscript in response to previous comments and included details about performance effects related to different overlap issues and t-test applications. However, I remain unconvinced by the results presented. The following comments need to be addressed.

Experimental design

1. The performance improvement may stem from both the t-test-based patch selection strategies and the modified CNN architecture. However, the author did not distinguish between these two factors. In Table 5, the performance of the "Proposed CNN model" is based on datasets selected through t-tests (as compared to metrics in Table 4). Were other models, such as ResNet-34 or Google Net, trained on the same dataset without using t-test selection?
2. I appreciate the author's reasoning for using a t-test to identify relevant patches. However, a simpler approach like thresholding could also effectively highlight the lung area, as its image intensity significantly differs from that of other regions.

Validity of the findings

3. Also, according to table 5, the proposed model trained on the full image had a better performance compared to the one trained with patches. This fact makes the train-on-patch strategy unnecessary.

---

## Round 0.3 · Minor Revisions

Minor Comments from reviewers:
1. Figure 2. Dimension labeling errors in CNN structure diagram. It should be 256 instead of 265.

Reviewer 3 ·

Basic reporting

Summary: This manuscript presents an automated diagnostic system for lung cancer detection from chest X-ray (CXR) images using a custom-designed convolutional neural network (CNN). The authors propose a novel preprocessing method involving dividing CXR images into overlapping patches and employing a t-test to retain informative patches, thus addressing dataset size limitations. The author revised the manuscript in response to previous comments and included details about new experiments. I appreciate the author’s effort revising the manuscript and included new experiments in response to previous comments. I suggest an acceptance.

Experimental design

NA

Validity of the findings

NA

Additional comments

Minor Comments:
1. Figure 2. Dimension labeling errors in CNN structure diagram. It should be 256 instead of 265.

---

## Round 0.4 · accepted · Accept

Reviewers are satisfied with the revisions, and I concur to recommend accepting this manuscript.